# Extracting particle size distribution from laser speckle with a physics-enhanced auto-correlation-based estimator (PEACE)

Qihang Zhang [1], Janaka C. Gamekkanda [2], Ajinkya Pandit [2], Wenlong Tang[3], Charles Papageorgiou [4], Chris Mitchell[4], Yihui Yang[4], Michael Schwaerzler [5], Tolutola Oyetunde[5], Richard D. Braatz [2], Allan S. Myerson [2] & George Barbastathis [6,7] ✉

Extracting quantitative information about highly scattering surfaces from an imaging system is challenging because the phase of the scattered light undergoes multiple folds upon propagation, resulting in complex speckle patterns. One specific application is the drying of wet powders in the pharmaceutical industry, where quantifying the particle size distribution (PSD) is of particular interest. A non-invasive and real-time monitoring probe in the drying process is required, but there is no suitable candidate for this purpose. In this report, we develop a theoretical relationship from the PSD to the speckle image and describe a physics-enhanced autocorrelation-based estimator (PEACE) machine learning algorithm for speckle analysis to measure the PSD of a powder surface. This method solves both the forward and inverse problems together and enjoys increased interpretability, since the machine learning approximator is regularized by the physical law.

Speckle results from the propagation of a wavefront whose phase has been strongly modulated by spatially variant features across a surface (or volume), so the speckle is an encoding of spatial patterns on the rough surface. As long as the morphology statistics are invariant, it is straightforward to relate statistical moments of the surface to the statistical moments of the speckle[1–3]. The laser speckle pattern has long been used to characterize surface roughness[3–12]. However, the method typically works only when the surface height fluctuation (equivalently, typical particle size) is smaller than or comparable to the light wavelength[3], limiting its application to surfaces encountered in many industrial processes, such as pharmaceuticals manufacturing. Electronic Speckle Pattern Interferometry (ESPI) can measure the surface motion distribution even at nanometer scales, but it is a two-step measurement with a reference beam, inhibiting real-time monitoring

and lacking absolute height information[13–15]. Laser speckle contrast imaging is qualitative and does not yield quantitative surface roughness[16]. Interferometric particle imaging can measure the particle size and shape, but it only works for a single particle or sparsely distributed particles[17–19].

Recent advances in Machine Learning have been successful in imaging through scattering media[20–25] and speckle suppression[26–31]. However, in both cases, the speckle pattern is treated as an unwanted disturbance. Extracting the scattering media information from the speckle has also been pursued qualitatively: for example, the classification of materials according to the scattered speckle patterns[5–8]. The main difficulty preventing further quantitative speckle analysis is the sensitivity of the phase signal to surface randomness, which hinders neural networks' ability to identify other underlying dynamics.

[1]Department of Electrical Engineering and Computer Science, Massachusetts Institute of Technology, Cambridge, MA 02139, USA. [2]Department of Chemical Engineering, Massachusetts Institute of Technology, Cambridge, MA 02139, USA. [3]Data Sciences Institutes, Takeda Pharmaceuticals International Co, 650 E Kendall St, Cambridge, MA 02142, USA. [4]Process Chemistry Development, Takeda Pharmaceuticals International Co, 40 Landsdowne St, Cambridge, MA 02139, USA. [5]Innovation and Technology Sciences, Takeda Pharmaceutical Company Limited, 200 Shire Way, Lexington, MA 02421, USA. [6]Department of Mechanical Engineering, Massachusetts Institute of Technology, Cambridge, MA 02139, USA. [7]Singapore-MIT Alliance for Research and Technology (SMART) Centre, 1 Create Way, Singapore 117543, Singapore. ✉e-mail: gbarb@mit.edu

Quantitative granularity characterization is desired in many applications[32–34], particularly the powder drying process in the pharmaceutical industry, during which the wet solid ("cake") is converted into a powder consisting of particles with the requisite size distribution. These powders are subsequently employed with other ingredients to form solid oral dosage forms such as tablets and capsules. However, agglomeration, deagglomeration, and crystal breakage are all likely during the drying process. Occurrence of hard agglomerates could influence content uniformity and functionality in the final drug product, e.g., if the active ingredient concentration becomes too high. Even though the parameters of the drying processes are generally well-controlled, the evolution of the particle sizes during agitation is not fully predictable. Thus, it is crucial to monitor particle sizes quantitatively in real-time and correct for abnormal size changes through feedback control on process parameters (e.g., temperature, agitation speed).

No real-time online monitoring methods exist presently, to our knowledge, that can detect early on and prevent such abnormal particle size changes for wet powder drying. Since the cake surfaces closely meet the Lambertian assumption[35,36], imaging by a standard camera from a distance compatible with the manufacturing setting (~0.2–0.5 m away from the powder) does not provide sufficient contrast or spatial resolution to extract the surface PSD. Machine vision to analyze the appearance of the cake surface and detect agglomerates is generally limited due to the same reason[37]. Imaging with cameras in situ relies on particle sparsity, which only works for solid suspensions rather than wet powder[38–40]. Moreover, this method is invasive, so there is a risk of the powder obscuring the viewing field and rendering the imaging operation impossible. Instead, manufacturers commonly rely on trained personnel to visually observe the mixing—but this can be subjective. Lastly, it is possible to extract a sample from the cake at fixed time intervals and pass it through a particle size analyzer instrument. However, this method is invasive and slow and, thus, not suitable for industrial use.

In this work, we propose a physics-enhanced autocorrelation-based estimator (PEACE) to extract the PSD of a powder surface from its laser speckle as shown in Fig. 1. With the help of the free-space propagation equations, we relate the ensemble-averaged spatial-integral autocorrelation function to the statistics of powder surface, i.e., the PSD. This relationship becomes the forward model for the estimator, yet it is inevitably incomplete. For example, particles may overlap along the longitudinal direction, which should not be introduced in the explicit model lest it becomes exceedingly complicated. Similarly, our experimental approach includes a finite spatial integral and temporal averaging of several frames, which are subject to sensor uncertainties in the model. Another limitation is that collecting sufficient experimental data to compensate for these uncertainties is prohibitively expensive.

We use PEACE to compensate for these uncertainties in the forward model. As shown in Fig. 1b, a small neural network called "generator" is combined with the forward model to form the final map from the PSDs to the experimentally collected autocorrelation images. In this way, starting from a modest amount of experimental data we can create a much larger synthetic (simulated) dataset. Finally, a deep neural network (DNN) called "estimator" is trained by the synthetic dataset to learn the inverse mapping from the speckle autocorrelation to the PSD. The estimator is overparameterized but generalizes well, confirming recent theoretical developments[41]. This method solves both the forward and inverse problems together, improving the machine learning model's generalization ability and interpretability. For example, the explicit forward model allows us to estimate bounds on our prediction ability. We note that the terms "generator" and "estimator" may be reminiscent of the generator and discriminator in Generative Adversarial Networks (GANs)[42] yet our approach is significantly different in that we do not employ adversarial training.

Here, we show that our method overcomes these limitations by providing a real-time, non-invasive, far-field optical probe (as shown in Fig. 1a) to monitor particle size distributions quantitatively. Especially for densely concentrated wet powders, this method is the first in-line measurement, and it is easily deployable in the industrial instrument.

## Results
### Forward model−speckle and particle statistics
With the help of the physics model, we derive the expression of the forward operator $H$ in the "Methods" section and in Supplementary

**a**

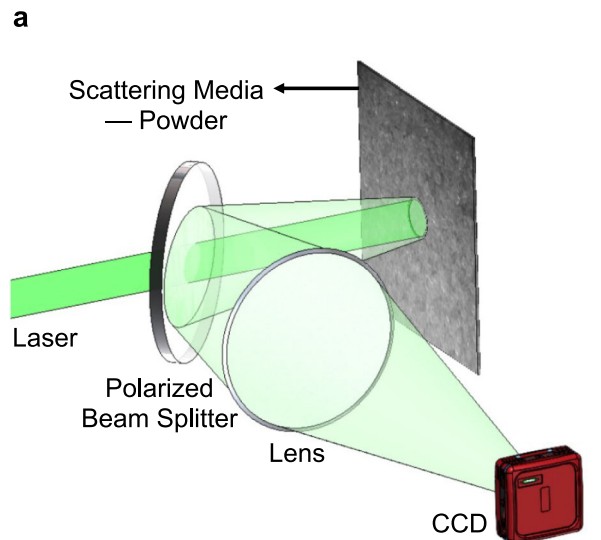

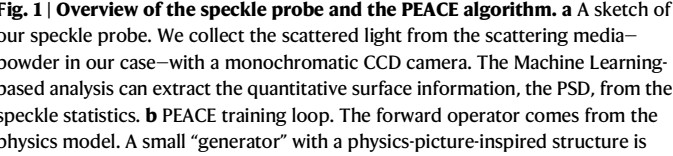

**b**

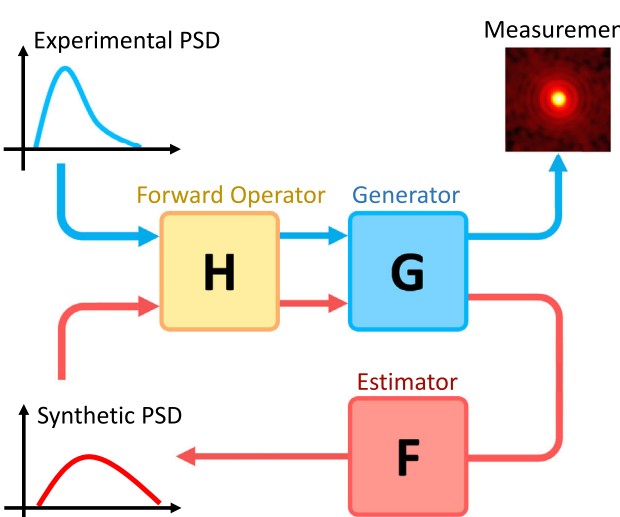

Fig. 1 | **Overview of the speckle probe and the PEACE algorithm. a** A sketch of our speckle probe. We collect the scattered light from the scattering media− powder in our case−with a monochromatic CCD camera. The Machine Learning-based analysis can extract the quantitative surface information, the PSD, from the speckle statistics. **b** PEACE training loop. The forward operator comes from the physics model. A small "generator" with a physics-picture-inspired structure is trained by a modest amount of the experimental data. The forward operator and the trained generator produce a much larger synthetic dataset. This synthetic dataset trains the DNN "estimator" to learn the mapping from the measured speckle autocorrelation to the particle size distribution (PSD). The 'generator' only contains 2.8k parameters, while the estimator has 377k parameters.

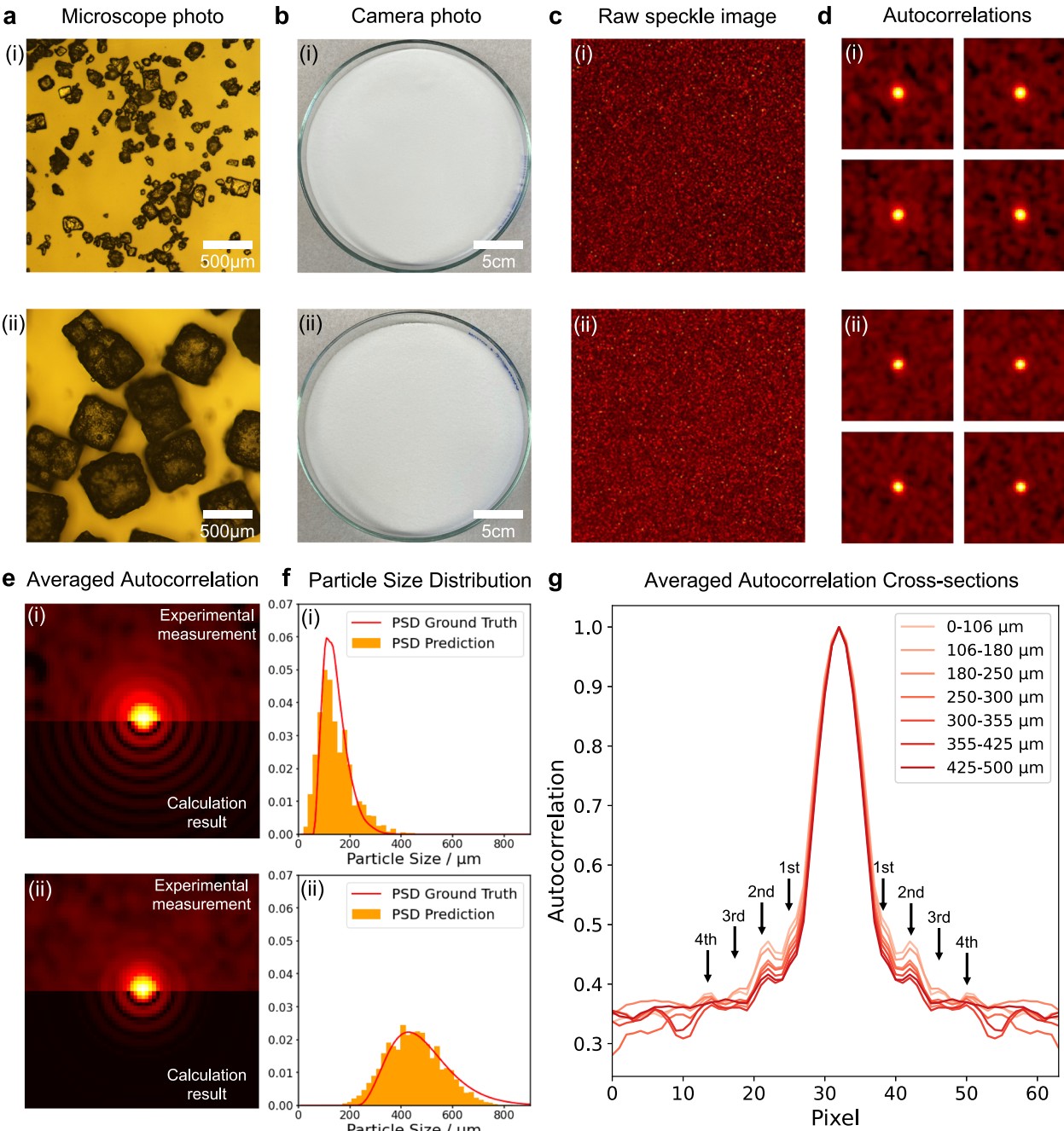

**Fig. 2 | KCl powder results with sizes (i) ~106–180 µm and (ii) ~425–500 µm.**
**a** Microscope and **b** commercial camera photos for two samples. **c** Raw speckle images collected by the CCD camera. **d** These four spatially integrated auto-correlation images are collected on the same sample with different particle positions. **e** The upper half image shows the averaged 1000 autocorrelation frames from the measurement. Since the particle position is ergodic, the temporal average equals the ensemble average of the autocorrelation image. The lower half image

shows the calculation result from the forward operator. **f** The ground truth particle size distribution (PSD) and the corresponding estimator prediction are plotted together. **g** Cross-section plots of the averaged autocorrelations in **e** for different samples with ascending PSDs. The positions of the high-order lobes are marked. We raise the autocorrelation to the power of one-eighth to enhance the sidelobes' visibility.

Section 3.

$$\langle A(u)\rangle = H(p(r)) = \frac{1}{C}\frac{4\sin^2\left(\frac{Du}{2}\right)}{D^2u^2}\left|\int p(r)\frac{\sin(ru)}{u}dr\right|^2 \quad (1)$$

where $\langle A \rangle$ denotes the ensemble average autocorrelation of the speckle pattern, $p(r)$ is the PSD in terms of number of particles, $D$ is the beam spot diameter which is much larger than the particle size $r$ in our range of interest (50–1000 µm), and

$C = \left|\int p(r)r\,dr\right|^2$ is the normalization factor. Therefore, the main ring-shape feature is dominated by $\frac{4\sin^2\left(\frac{Du}{2}\right)}{D^2u^2}$, and the side-lobe intensities are modulated by $\left|\int p(r)\frac{\sin(ru)}{u}dr\right|^2$. We design the progression of a typical experiment based on this forward operator as shown in Fig. 2.

Figure 2a shows two sample sets with different particle sizes imaged through a regular microscope. Figure 2b are photos collected

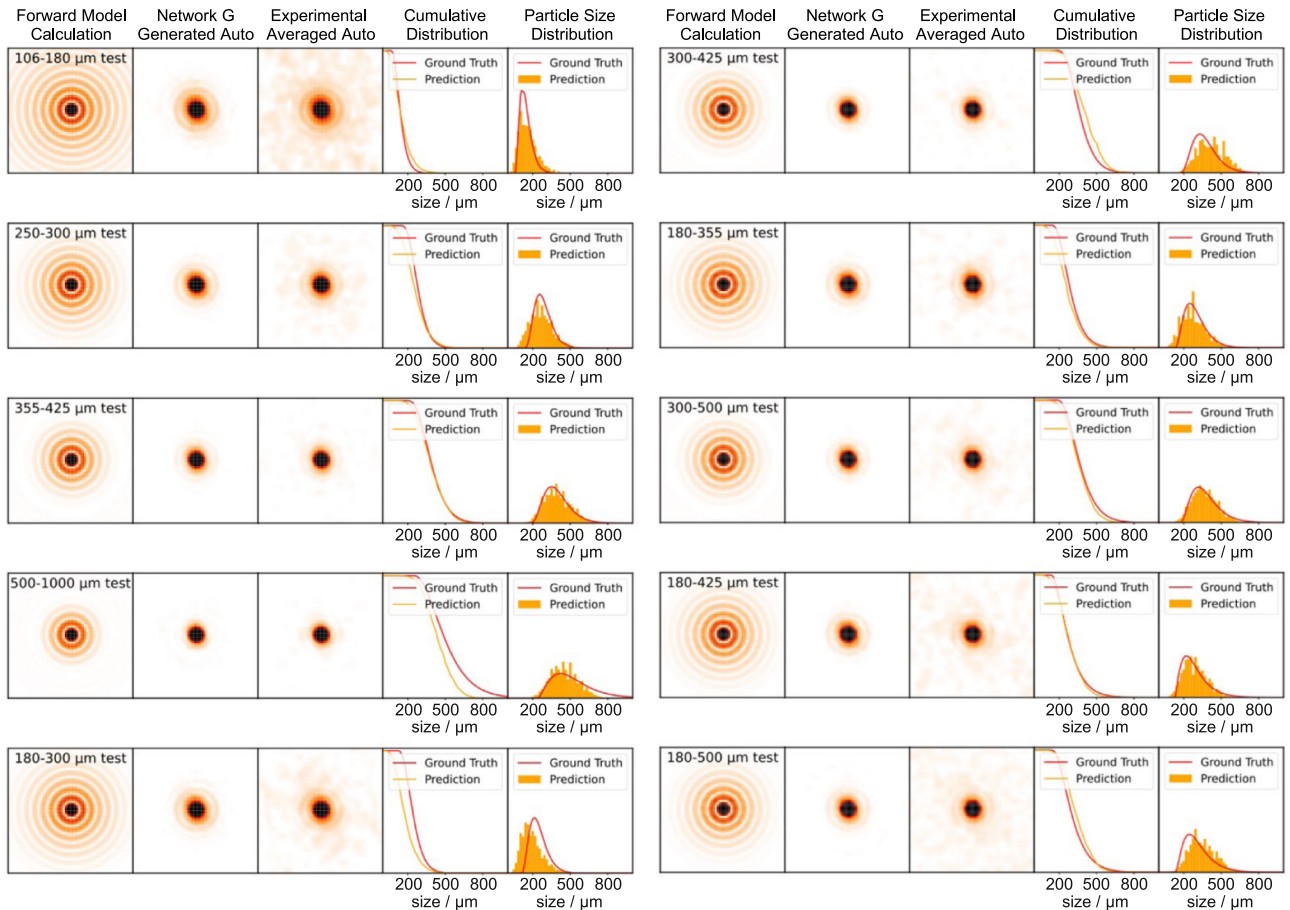

**Fig. 3 | Generator and estimator prediction results with the experimental test dataset.** The results of 10 test sample sets are plotted. The first and second columns show the output from the forward operator H and the generator G, respectively. The measured 200 frames averaged autocorrelations are plotted in the third column. The prediction (marked as orange) from the estimator F and ground truth (marked as red) of the cumulative distributions and the corresponding PSDs are plotted together in columns 4 and 5.

by a commercial camera, with broadband and spatially incoherent illumination. The contrast and resolution are too coarse to resolve the particles, not displaying any discernible features that could be attributed to particle size. Figure 2c shows their raw speckle images collected from the CCD camera. Figure 2d shows the respective speckle autocorrelation patterns, which are subsequently averaged as shown in Fig. 2e. The calculation results in Fig. 2e are passed to a two-stage neural network consisting of the generator and the estimator. The generator corrects the physical model for uncertainties, such as overlapping particles that would be very hard to model analytically. The estimator receives its input from either the generator or the measured results (shown in Fig. 2e) and produces the cumulative distribution of the particle sizes. By one more differentiation step, we obtain the final estimate of the PSD as a vector of powder concentrations *vs.* particle size. Figure 2f shows one typical test result from a particle distribution never shown during the neural network's training stage, as computed by the combined generator-estimator algorithm. This estimated PSD can be compared with the ground truth PSD that we established using a commercial particle size analyzer, the "Mastersizer" (details in Supplementary Section 2). The complete details of this process are in the Supplementary Sections 3 and 4. Figure 2g shows cross-sections of the averaged autocorrelation for different sample particle sizes. The intensities of the first and second-order lobes monotonically decrease with ascending PSDs. The higher-order lobes are merged into the background fluctuations for the large particle size sample, while for the small particle size sample the lobes are clearly resolvable.

## Test results and model visualization for PEACE

We also conducted a thorough analysis of the generalization ability of the algorithm, shown in Fig. 3. This experimental test dataset was disjoint from the training data. The first column is the calculation result for the measured PSD from the forward model. The second column shows the image produced by the generator, while the third column shows the corresponding ground truth, i.e., the experimental averaged autocorrelation. These images are averaged from 200 autocorrelation images, different from the 1000 frames averaged one in Fig. 2e. Reducing the number of averaging frames sacrifices the signal-to-noise ratio but speeds up the data collection time for each prediction. There is still a slight mismatch between the generated image and the measured image in the surrounding area. These noise shape features are the residual from the ensemble average because the image is averaged with a finite number of frames only. Since they are away from the region of interest (later discussed in Fig. 4e), this deviation will not affect the performance of the estimator. The predictions and the ground truths measured from the particle size analyzer are plotted together in columns 4 and 5. The cumulative distribution follows from a clear physical definition of the size distribution for the non-spherical particles, such as the cubic-shaped KCl particles shown in Fig. 2a. More details about the cumulative distribution are included in Supplementary Section 4.

We use two methods to visualize our estimator. The first is plotting the output of each stage (the estimator's detailed structure is in Supplementary Section 4). As shown in Fig. 4a–d, the first stage output deemphasizes the zero-order peak and differentiates the side region.

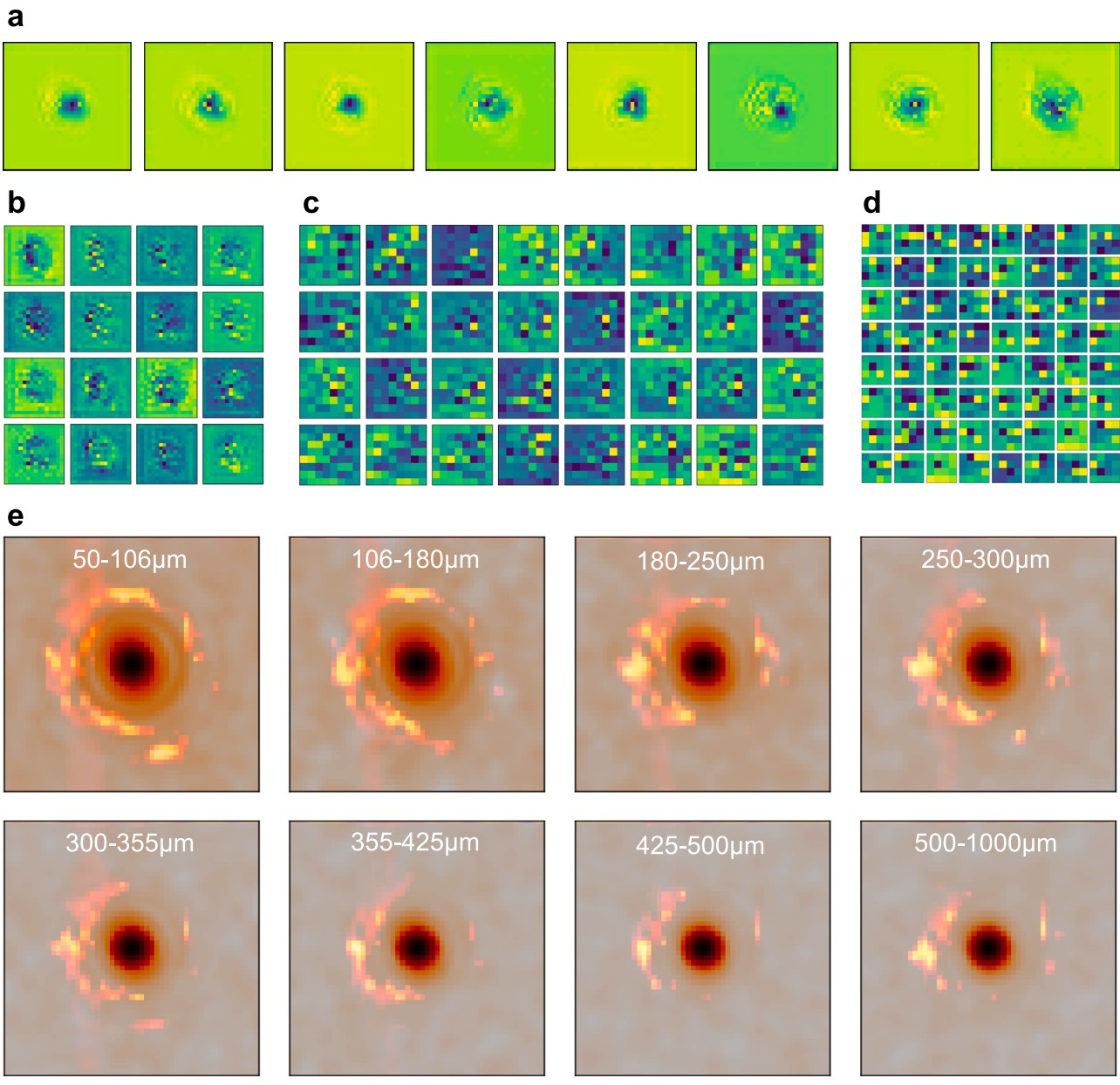

**Fig. 4 | Network visualization of the estimator F. a** The output of the first stage with 8 filters. **b** The output of the second stage with 16 filters. **c** The output of the third stage with 32 filters. **d** The output of the fourth stage with 64 filters. **e** GradCAM results for different sample sets. The input images are plotted together with the "importance" maps which consist of the flame shape features.

The second stage enhances the differentiation in the side area. The last two stages decode the information and extract the features.

The other method is the Gradient-weighted Class Activation Mapping (GradCAM)[43]. This method visualizes the regions of input that are "important" for predictions. The results for different samples are plotted in Fig. 4e. For small-size samples, the attention is spread over the high-order lobes. For the big-size samples, the attention is concentrated on the second-order lobes since the higher-order lobes are no longer distinguishable. The attention for all plots avoids the central peak region, which is consistent with our theory.

**Time-lapse PSD monitoring in the drying process**

To validate the real-time monitoring applicability of our method, we carried out a time-lapse PSD measurement of an entire filter drying process[44]. The detailed information of our dryer is included in Supplementary Section 1. This demo process operated on 280 g of KCl powder with a mixed solvent (water 40 g/Ethanol 60 g). Throughout

the process, we maintained conditions of temperature 26 °C, pressure −720 mbar and agitation speed 4 rpm. Figure 5b shows the PSD map *vs.* time. The sampling period of the PSD measurement was 15 s, including the data collection time for 200 frames and the computation time. The computer used for this measurement was an Intel Xeon W2245 CPU, 64 GB RAM, and NVIDIA Quadro RTX 5000 GPU with 64 GB VRAM.

For the duration of 5 to 25 min since the beginning of the process, we observed a gradual size increase compared to the original PSD. Since no crystallization or crystal growth could take place during drying, we think soft agglomeration instead occurred. From 25 to 40 min, the PSD gradually decreased to the original distribution, which is indicative of deagglomeration. At the far end (>75 mins), crystal breakage[45] caused the size distribution to decrease slightly compared to the original. PSD curves sampled at different times are plotted in Fig. 5a, together with Mastersizer results serving as ground truth for the beginning and ending time. They match well and the slight crystal breakage is clearly resolved for both Mastersizer and speckle

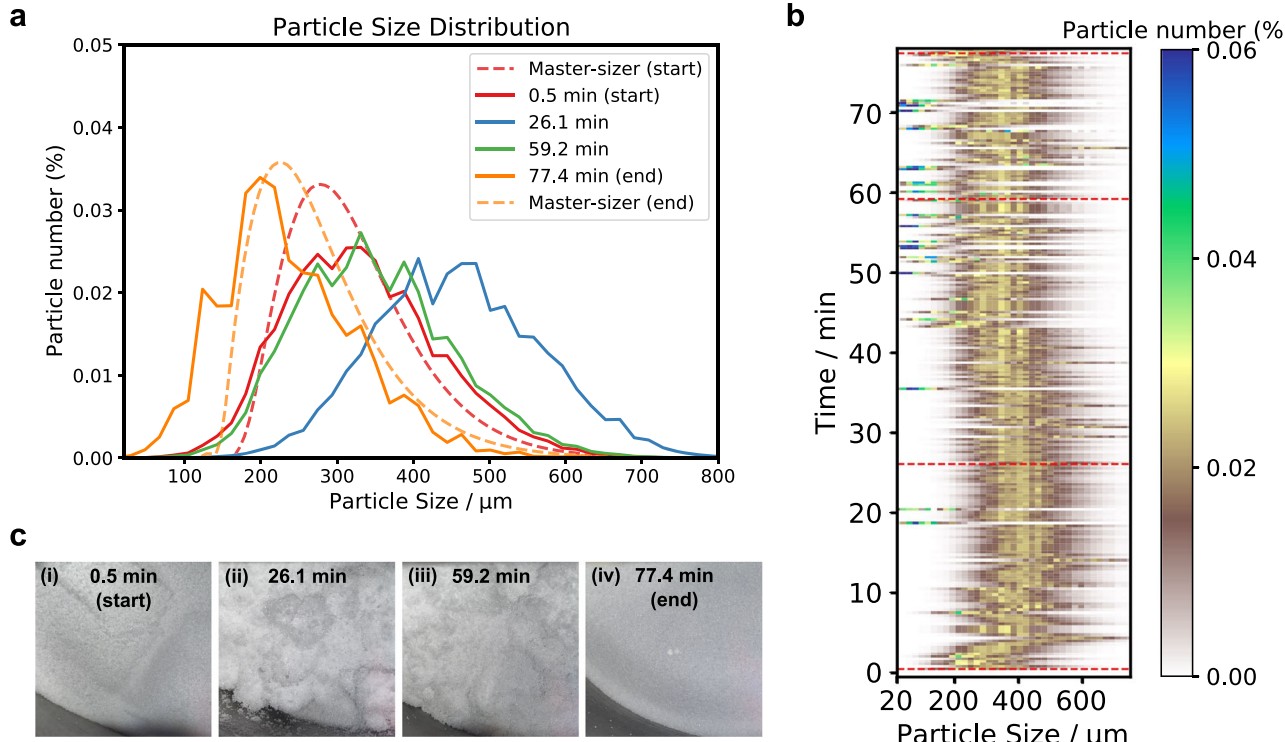

**Fig. 5 | Time-lapse PSD measurement in the drying process. a** Measured PSDs at a few selected time points corresponding to the dashed red line marks in (**b**). The dashed line plots are the Mastersizer measurements serving as ground truth at the beginning and the end of the process. **b** Time-lapse PSD map during the drying process. We observed soft agglomeration occurring from 5 to 25 min, and deagglomeration from 25 to 40 min. **c** Corresponding camera photos at the times where the PSDs were sampled in (**a**). These photos were taken from the same optical window as the laser beam using a commercial iPhone 11.

predictions. The PSD curve shifts toward the right from the start time to 26.1 min, and then decreases backwards to the original position as plotted for the next 59.2 min. We did not measure the PSD with the Mastersizer when the powder was wet because in this process soft agglomerates were too fragile to allow any mechanical contact. This is one of the advantages of our non-invasive measurement compared to the traditional way with the Mastersizer. Figure 5c are camera photos, taken with an iPhone 11, corresponding to the marked times in (a) and (b) for reference. (i) and (iv) are dry powders, while the (ii) and (iii) are wet powder images. It is hard to distinguish different sizes from macroscopic textures.

## Discussion
We have decoded the size information in the speckle pattern quantitatively with the help of the physics-enhanced autocorrelation-based estimator (PEACE). The data flow goes through a sequence of two neural networks, the generator and estimator, both informed by the law that we discovered relating the PSD to autocorrelation side lobes. Physics is involved in this method in four aspects. (1) The theoretical model guides the preprocessing of the raw speckle image, which is the ensemble-averaged autocorrelation (Supplementary Section 3). (2) The forward operator joins the PEACE training loop explicitly. (3) The generator's structure is inspired by the physics picture, resulting in high performance with only a few parameters, which is crucial for avoiding overfitting (Supplementary Section 4). (4) The output of the estimator is the cumulative size distribution, which is applicable to different particle shapes. Test results on needle shape materials, along with a theoretical discussion of the relationship between non-spherical particles and cumulative distributions are included in Supplementary Section 4.

One advantage of this physics-informed strategy is interpretability. The activation map results match our forward model very well, which means the size information is stored in the side lobes, and that

enables an evaluation of the performance of the neural network in addition to the loss metric. For example, we want the generator's output to better match the ground truth at the 2nd–5th lobes region. The center and the surrounding area are not very important. It is also possible to estimate the bound of the prediction ability from the forward model. Based on our theory, we even know how to push the prediction range into other regions of interest by tuning the parameters of the optical system. This is further discussed in Supplementary Section 5.

Our estimator fails to predict the double-peak shape PSD in a stress test with the synthetic data, yet it is able to estimate the appearance of large particles from the continuous monitoring—which is the most important for process control. This observation can be explained with the forward model; the results and explanation of the stress test are in Supplementary Section 6. This PEACE method is well suited for the extension to other contexts, including processes in the pharmaceutical industry such as blending and milling.

## Methods
### Optics apparatus
Our far-field optical probe can easily combine with an agitated filter dryer (AFD) typical of those used in the pharmaceutical industry. The wet solid is sealed in the dryer and the laser beam is delivered through a glass window. Figure S2a is a picture of the optics apparatus as we implemented it. For safety, the entire beam path except the output port facing the window is enclosed in an optical cage and optical tubes to prevent scattered light from escaping. This structure is compact, portable and, therefore, easy to transfer among different dryer systems without realignment. An agitator placed along the axis of the container impels the sample at a rotation speed of 4 rpm. The laser operates at 532 nm wavelength, and it is fairly easy to replace the source with a different wavelength, if desired. The angle of incidence on the surface is chosen to be approximately

10°, so as to avoid specular back-reflection from the window onto the camera.

The optical beam path is shown in Fig. S2b. The laser beam is expanded to 4.8 mm with a beam expander in the telescope configuration, consisting of lenses L1 (focal length 25 mm) and L2 (30 mm). The beam is initially polarized in the direction parallel to the plane shown in the diagram, and so it is reflected by the polarizing beam splitter (PBS) to reach the sample inside the dryer through the glass window. Passage twice through the quarter-wave plate rotates the outgoing polarization direction from parallel to perpendicular so that the scattered light from the potassium chloride (KCl) powder sample is now transmitted through the PBS and propagates vertically upwards. The wave plate is also tilted by approximately 10 degrees for the same reason as the beam, to minimize specular back-reflection. Lens L3 (250 mm) concentrates the scattered light so that the CCD (model ZWO ASI183MM Pro) may capture an angular range as extensively as possible. More information about our apparatus can be found in Supplementary Section 1.

## Theoretical derivation of the Forward Model

The theory of the forward problem is established to link from a particular particle distribution to the raw speckle. The formulation of the forward problem is necessarily stochastic, treating the PSD as a probability density function which, in turn, determines the ensemble autocorrelation function of the raw speckle. The sketch for the initial simple analytical model is shown in Fig. S2c. We assume that only particles can scatter the light. Without loss of generality, the reflectivity is set to 1. In other words, the sketch considers the particles as a thin optical mask, which we denote as $a(x)$.

We interpret the particle radius $r_i$ as a random variable distributed according to the PSD $p(r)$. The particle location $x_i$ is also a random variable uniformly distributed across the object plane. $H(x)$ describes the surface height resulting from the randomly placed particles. The corresponding phase of the scattered light is $w(x) = \exp(j\frac{2\pi}{\lambda}H(x))$. In the Fourier plane of L3, the electric field $E(x)$ is

$$E(x) = \int e^{j\frac{2\pi}{\lambda f_3}x\xi}S(\xi)d\xi, \tag{2}$$

where

$$S(x) = a(x)w(x), \tag{3}$$

and $f_3$ is the focal length of L3. The intensity collected by the CCD camera is

$$I(x) = |E(x)|^2 = \iint e^{j\frac{2\pi}{\lambda f_3}x(\xi_1-\xi_2)}S(\xi_1)S^*(\xi_2)d\xi_1 d\xi_2. \tag{4}$$

We now define the spatial-integral autocorrelation of the speckle image as

$$A(u') = \int I(x)I(x+u')dx. \tag{5}$$

This equation may be rewritten in the form

$$A(u) = |\sum_i \frac{\sin(r_i u)}{u}e^{j2\pi x_i u}|^2. \tag{6}$$

Here, $u = \frac{u'}{\lambda f_3}$, $i$ is the index of the $i$th particle, and $r_i$ and $x_i$ are the radius and the position for the $i$th particle, respectively. The full derivation leading from (5) to (6) is in Supplementary Section 3, equations (M1)–(M14). If a sufficient number of particles find themselves within

the field of view, then Eq. (6) can be reformulated as

$$A(u) = |\int dr p(r)\frac{\sin(ru)}{u}\sum_i e^{j2\pi x_i u}|^2. \tag{7}$$

This expression corresponds to the images in Fig. 2d. The granular feature results from the term $\sum_i e^{j2\pi x_i u}$ rather than measurement noise. The position information is encoded in this summation term. Since the particle coordinate $x$ is immaterial, we may eliminate it by ensemble-averaging the autocorrelation $A(u)$. Starting from Eq. (7), $\langle A(u)\rangle$ becomes

$$\langle A(u)\rangle = \frac{4\sin^2\left(\frac{Du}{2}\right)}{D^2 u^2}|\int p(r)\frac{\sin(ru)}{u}dr|^2. \tag{8}$$

The details are in the Supplementary Section 3. In (8), $\langle\cdot\rangle$ denotes the ensemble average and $D$ is the beam spot diameter. This average operation cannot suppress the granular feature directly, but our physics model reveals that the $\sum_i e^{j2\pi x_i u}$ term is averaged into the $\frac{4\sin^2\left(\frac{Du}{2}\right)}{D^2 u^2}$ term, which is irrelevant to the specific position distribution and only depends on the beam spot size. In this way, we change the stochastic expression (6) into a deterministic expression (8) for a given PSD, which ensures that the supervised learning is able to capture the correct map from the calculation result to the experimental result. Equation (8) is an intuitive yet approximate forward model between the raw speckle images and the PSD $p(r)$ through the averaged speckle autocorrelation function $\langle A(u)\rangle$.

## Data availability

All processed data in this study have been deposited in the Harvard Dataverse under accession https://dataverse.harvard.edu/dataset.xhtml?persistentId=doi:10.7910/DVN/FZUG9V. The raw data are too big to upload into the public repository, please contact the author to request them. Line plots in Figs. 2 and 5 are provided in the Source Data file. Source data are provided with this paper.

## Code availability

The entire original code has been deposited at https://github.com/qhzhang95/PEACE_Speckle and is publicly available (https://doi.org/10.5281/zenodo.7497506).

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

## Acknowledgements

The authors acknowledge the MIT SuperCloud for providing HPC resources that have contributed to the research results reported within this paper/report. This research was supported by Millennium Pharmaceuticals, Inc. (a subsidiary of Takeda Pharmaceuticals), grant No. D824/MT15 (A.M., G.B., and R.B.).

## Author contributions

G.B. and Q.Z. conceived the project and developed the theory. Q.Z. and J.G. designed the optical experiment and collected the data. Q.Z. analyzed the speckle data and developed the algorithm. J.G., A.P., Q.Z., W.T., C.P., C.M., and Y.Y. designed the filter dryer set up. Q.Z. and A.P. carried out the time-lapsed PSD drying experiment. Q.Z., J.G., and G.B. prepared the original manuscript and W.T. helped in the manuscript writing. C.P., C.M., Y.Y., M.S., and T.O. also contributed to the manuscript. G.B., A.M., and R.B. supervised this project.

## Competing interests

The authors declare the following competing interests. Patent applicant: Massachusetts Institute of Technology. Name of inventor(s): George Barbastathis, Qihang Zhang, Janaka C Gamekkanda Gamaethige, Richard D Braatz, Allan S Myerson. Application number: PCT/US22/

50045. Status of application: Pending. Specific aspect of manuscript covered in patent application: System and method for determination of particle size distributions. There are no other competing interests.
