## [Peer Review File · Nature Communications]

Extracting Particle Size Distribution from Laser Speckle with A Physics-Enhanced AutoCorrelation-based Estimator (PEACE)REVIEWER COMMENTS

Reviewer #1 (Remarks to the Author):

The submitted paper deals with in-line particle size distribution estimation using laser backscattering from KCl powder of interest.

It is a well written manuscript with a clear objective, however, there are some questions that should be answered before the acceptance.

There are many questions about the applicability of the method.

Probably, powder on the observation window might cause significant differences in the laser speckle, thus the applicability of this technique is questionable when more realistic APIs with lower particle sizes are used. How the observation window is cleaned during the process?

1. Why agglomerates of 700 micron are critical for KCl? Is it possible to differentiate the aggregates from the single crystals? I feel that it is not possible right now. Larger crystals can be detected only not the aggregates. For the detection of the aggregation of two particles “particle tracking” would be needed which is impossible with this “laser speckle-based technique”.

In the experimental part no aggregated particles were detected only sieved KCl. I think it is better to focus on the PSD prediction instead of aggregate detection in the manuscript.

2. Computer vision with a standard industrial camera equipped with an appropriate objective should work well if visual observation by trained personnel is used right now in the production plants to detect the aggregation (line 93). Particle size of 700 micron is large. There are many objectives with appropriate magnification to detect particles over 100 micron. I also work with KCl and I have photos about KCl crystals (DV10 = 26 micron, DV50 = 186 micron, DV90 = 402 micron) with a 35 mm focal length objective (<https://pyramidimaging.com/Basler-Lens-C11-3520-12M-P-f35mm---Lens-p-73256.aspx>). (Working distance was about 150-200 mm).

A microscope image about the same material:

3. Does it work with needle-like crystals as well? Many types of aggregation can occur. It is very time consuming to “teach” new models for the all the new materials. Aggregated particles also should be shown to the model during the learning phase as well.
4. If the crystal morphology is changed during the crystallization the laser speckle will change as well.

Reviewer #2 (Remarks to the Author):

Zhang et al. describe a novel quantitative analysis method that uses back-scattered speckle images to estimate particle size distribution (PSD). This method can be applied to monitor powder size distribution in the pharmaceutical industry. In the analysis, a physical forward model based on scattered field data is used together with partial experimental data in a deep neural network to generate further synthetic data and estimate PSD with an estimator to solve the inverse problem. In general, the research problem in the paper is interesting and solid, in the sense that extracting quantitative information from scattered fields is not an easy task in two aspects: understanding the image formation process and solving parameter estimation inverse problems in a generalized way that is applicable to real-life situations. In order to solve both aspects of the problem, the authors incorporated physics-informed machine learning. In addition, the paper is overall carefully and rigorously written. I particularly appreciate the way in which the limitations and weaknesses of the method are explained clearly and honestly. However, I still have several major concerns regarding the paper that need to be addressed before I consider it for acceptance:

Major comments:

- 1) An important application of the work is the monitoring of the wet powder drying process in a "real-time, non-invasive" manner. The data and setup demonstrated so far only support the non-invasive claim. The real-time claim, which should be one of the most exciting claims, does not appear to be supported by any obvious data:

1.1) During the drying of wet powder, I would expect dynamic variations in speckle images that may decorrelate over time, so the ensemble average model may not be applicable. There is a need for the authors to provide more technical details regarding the imaging and data processing time in comparison with the speckle decorrelation time in the typical wet powder drying process. To support the real-time claim, I would expect a time-lapsed measurement of PSD.

1.2) It also appears that the model does not account for dynamics as well. Does the model assume that each measurement is made within the period of speckle correlation?

1.3) Although the authors provided a ground truth comparison study, it was done with fixed powder rather than wet drying powder as claimed in the introduction. Is there any experiment to validate the "real-time" measurement component? It is necessary for the authors to provide more image data from different times, as well as its analysis during the validation experiment with a known ground truth PSD.

Otherwise, this claim should be tuned down and another discussion section should be added to the paper.

2) As the data analysis method in the paper has only quantified particle sizes of a few hundred microns, which is within the capability of a typical wide-field microscope with a large field of view design with a few hundred microns of resolution. What is the clear technical advantage of this method over direct microscopic imaging?

3) Codes for the model and DNN should be publicly available for easy reproducibility of the work.

4) In lines 38-39, "The laser speckle pattern has long been used to characterize surface roughness. ", but there are not enough references cited and attributed, please cite the work Buchanan, J., Cowburn, R., Jausovec, AV. et al. Nature 436, 475 (2005) and other significant previous work.

Minor comments:

1) According to equation M15, is the ensemble average the average over time? Please clarify in notation which parameter in $A(u)$ is averaged from equation M14 to M15. If it is time or the number of measurements, then M14 should include the parameter t , i.e. If it is time or the number of measurements, then M15 should be $A(u) = \langle A(u,t) \rangle |t$. In the paper, it is also not stated clearly how many average times have been calculated for each experiment. Please clarify.

2) The setup description in the Method part should include more information, such as the focal lengths of the lenses, camera information, etc.

RESPONSE LETTER TO REVIEWER COMMENTS

We thank the Reviewers for their time and for their valuable comments. We hope that the Reviewers will agree with us that their criticism has resulted in a significantly improved manuscript. We have addressed all the comments with special emphasis on the two topics emerging from the reviews as most salient; namely, “the applicability of the method” (from reviewer #1) and “the validation of real-time monitoring” (from reviewer #2).

Reviewer #1 (Remarks to the Author):

The submitted paper deals with in-line particle size distribution estimation using laser backscattering from KCl powder of interest. It is a well written manuscript with a clear objective, however, there are some questions that should be answered before the acceptance. There are many questions about the applicability of the method.

We thank the Reviewer for the helpful questions relating our work to the real-life application. We have conducted additional experiments to validate the applicability of our method, and have included them in the main text and the supplementary materials. Point-to-point responses are below.

1. Probably, powder on the observation window might cause significant differences in the laser speckle, thus the applicability of this technique is questionable when more realistic APIs with lower particle sizes are used. How the observation window is cleaned during the process?

We appreciate this question. Although, as we elaborate in points (1-2) below, we have not encountered this problem yet, in points (3-4) we describe a few methods for cleaning up the window during the process.

(1) We agree that the powder on the observation window will affect the raw laser speckle pattern. Fortunately, few individual particles will not affect much the ensemble averaged autocorrelations and the corresponding particle size distribution. This is because these are statistical properties of the powder surface in its entirety and, hence, resilient to small perturbations. According to our forward model,

$$\langle A(u) \rangle = H(p(r)) = \frac{4 \sin^2\left(\frac{Du}{2}\right)}{D^2 u^2} \left| \int p(r) \frac{\sin(ru)}{u} dr \right|^2,$$

as long as the density of particles on the window is much smaller than the powder density in the API, the $p(r)$ term will still be dominated by the powder in the API.

(2) For our current experiment, the lowest size that we can detect accurately is around 50 μm (revised Sup. line 138 & 260, also added in the main text line 106). For powder sizes above 50 μm , we observed minor window contamination during the drying process, not affecting our measurement. For example, the following picture shows the worst case for window contamination, and this is within the scope of “few individual particles” referred to in point (1). It is not necessary to clean the window during the process for our current apparatus. More details about our time-lapsed PSD measurement in the drying process are in the response to the question 2 below.

(3) For many industrial-level dryers, the vertical distance between the window and the powder surface could reach more than 1 meter. Gravity mitigates window contamination even more since the particles do not have enough mechanical energy to reach the height of the window above them.

(4) We could apply certain accessories on the window to prevent window contamination. For example, we can purge a gas flow around the window to blow away the particles (<https://www.jmcanty.com/product/jet-spray-ring-2/>). Moreover, we could heat up the window to prevent the solvent condensation (<https://www.ljstar.com/product-line/heated-sight-glass-metaclamp-for-fog-free-viewing/>). With these strategies, we are confident that the window during the drying process would maintain sufficient optical quality for our measurement.

2. Why agglomerates of 700 micron are critical for KCl?

We appreciate the Reviewer's careful scrutiny of our language use in this issue. We use KCl as an exemplar for our concept proof, but 700 μm particle size is not special for KCl. Our original aim for the statement in line 175 is to indicate that our method reliably detects big size particles and this is critical for the application purpose (where of course the particles would not be KCl). Our original statement in line 175,

“it is able to estimate the appearance of agglomerates exceeding a critical size around 700 μm from the continuous monitoring –which is the most important for process control.”

has been revised in line 195 as

“it is able to estimate the appearance of large particles from the continuous monitoring –which is the most important for process control.”.

Is it possible to differentiate the aggregates from the single crystals? I feel that it is not possible right now. Larger crystals can be detected only not the aggregates. For the detection of the aggregation of two particles “particle tracking” would be needed which is impossible with this “laser speckle-based technique”. In the experimental part no aggregated particles were detected only sieved KCl.

We thank the Reviewer for pointing out the difference between the aggregates and the single crystals. This comment has been particularly helpful to us for articulating the utility of our method.

We acknowledge that this method cannot discriminate between aggregates and single crystals from a single particle size distribution (PSD) measurement. In addition, we would like to point out that we could access this information indirectly from continuous PSD monitoring combined with certain prior knowledge on the process. Specifically for the drying process, agglomeration, deagglomeration and crystal breakage are all plausible. We carried out a time-lapse experiment for the drying process, with the result shown in the revised manuscript in Fig. 5 (the same experiment was used to address some questions by Reviewer 2). From 5 min to 25 min, we observed a gradual size increase compared to the original PSD. Since there is no crystallization or crystal growth during drying, we think that what we observed was soft agglomeration. From 25 min to 40 min, the PSD gradually decreased to the original distribution, which we think is evidence of deagglomeration. At the ending point (>75 mins), crystal breakage occurred so that the size distribution slightly decreased compared to the original one. This experiment validates that we may use the dynamics of the PSD together with process knowledge to infer the aggregates, even though we cannot perform “particle tracking” *per se* (and, arguably, it is not necessary.)

The details about this experiment have been added in the main text as the following quotation (lines 152-173):

“To validate the real-time monitoring applicability of our method, we carried out a time-lapse PSD measurement of an entire filter drying process⁴¹. The detailed information of our dryer is included in Supplementary Section 1. This demo process operated on 280g of KCl powder with a mixed solvent (water 40g/Ethanol 60g). Throughout the process, we maintained conditions of temperature 26 °C, pressure -720 mbar and agitation speed 4 rpm. Fig. 5b shows the PSD map vs. time. The sampling period of the PSD measurement was 15 seconds, including the data collection time for 200 frames and the computation time. The computer used for this measurement was an Intel Xeon W2245 CPU, 64 GB RAM, and NVIDIA Quadro RTX 5000 GPU with 64 GB VRAM.

Fig. 5 | Time-lapse PSD measurement in the drying process. (a) Measured PSDs at a few selected time points corresponding to the dashed red line marks in (b). The dashed line plots are the Mastersizer measurements serving as ground truth at the beginning and the end of the process. (b) Time-lapse PSD map during the drying process. We observed soft agglomeration occurring from 5-25 min, and deagglomeration from 25-40 min. (c) Corresponding camera photos at the times where the PSDs were sampled in (a). These photos were taken from the same optical window as the laser beam using a commercial iPhone 11.

For the duration 5 to 25 minutes since the beginning of the process, we observed a gradual size increase compared to the original PSD. Since no crystallization or crystal growth could take place during drying, we think soft agglomeration instead occurred. From 25 to 40 minutes, the PSD gradually decreased to the original distribution, which is indicative of deagglomeration. At the far

end (>75 mins), crystal breakage⁴² caused the size distribution to slightly decrease compared to the original. PSD curves sampled at different times are plotted in Fig. 5a, together with Mastersizer results serving as ground truth for the beginning and ending time. They match well and the slight crystal breakage is clearly resolved for both Mastersizer and speckle predictions. The PSD curve shifts toward the right from the start time to 26.1 minutes, and then decreases backwards to the original position as plotted for the next 59.2 min. We did not measure the PSD with the Mastersizer when the powder was wet because in this process soft agglomerates were too fragile to allow any mechanical contact. This is one of the advantages of our non-invasive measurement compared to the traditional way with the Mastersizer. Fig. 5c are camera photos, taken with an iPhone 11, corresponding to the marked times in (a) and (b) for reference. (i) and (iv) are dry powders, while the (ii) and (iii) are wet powder images. It is hard to distinguish different sizes from the macroscopic textures.”

I think it is better to focus on the PSD prediction instead of aggregate detection in the manuscript.

We thank the Reviewer for this suggestion. We have made corresponding changes in our introduction to indeed emphasize PSD prediction over aggregate detection, as follows (line 74-85):

Original:

“A particular application of our method is the powder drying process in the pharmaceutical industry, during which the wet solid (“cake”) is converted into a powder consisting of particles with the requisite size distribution. These powders are subsequently employed with other ingredients to form solid oral dosage forms such as tablets and capsules. However, the formation of aggregates during the drying process is challenging because of “hard agglomerates” (crystals bonded together) which can influence content uniformity and functionality in the final drug product, e.g., too much active ingredient. Even though the parameters of the drying processes are generally well-controlled, the evolution of the particle sizes during agitation is not fully predictable. The breaking of hard aggregates is possible in some cases, but delumping the powder requires additional equipment and

time, resulting in added cost to the process and loss of products. Thus, it is crucial to monitor particle sizes quantitatively in real-time, to detect the beginnings of agglomeration as early as possible and correct through feedback control on process parameters (e.g., temperature, agitation speed).

*No real-time online monitoring methods exist presently, to our knowledge, that can detect early on and prevent such **abnormally large agglomerates** for wet powder drying.....”*

Rephrased:

*“A particular application of our method is the powder drying process in the pharmaceutical industry, during which the wet solid (“cake”) is converted into a powder consisting of particles with the requisite size distribution. These powders are subsequently employed with other ingredients to form solid oral dosage forms such as tablets and capsules. **However, agglomeration, deagglomeration and crystal breakage are all likely during the drying process. Occurrence of hard agglomerates could influence content uniformity and functionality in the final drug product, e.g., if the active ingredient concentration becomes too high.** Even though the parameters of the drying processes are generally well-controlled, the evolution of the particle sizes during agitation is not fully predictable. Thus, it is crucial to monitor particle sizes quantitatively in real-time and **correct for abnormal size changes** through feedback control on process parameters (e.g., temperature, agitation speed).*

*No real-time online monitoring methods exist presently, to our knowledge, that can detect early on and prevent such **abnormal particle size changes** for wet powder drying.....”*

3. Computer vision with a standard industrial camera equipped with an appropriate objective should work well if visual observation by trained personnel is used right now in the production plants to detect the aggregation (line 93).

We thank the Reviewer for this point, which was salient also during our internal discussions while formulating this project.

Compared to computer vision with a commercial camera, our method infers particle size information from much smaller feature length scales. An experienced operator could make a rough guess (without any guarantee on the accuracy) based on the surface texture during the drying process, and further check the PSD invasively with a master-sizer instrument (*i.e.*, a particle size analyzer.) While detailed psychophysical analyses of how these operators make decisions are not available (to our knowledge), we speculate that humans assess the hue and texture in the back-scattered white light to assess the state of the surface. They are also aided by ample experience. We agree that a computer-vision based method with a commercial camera might succeed in emulating this human function (which our technique does not do either—our technique relies on the statistics of coherent back-scatter instead) without resolving the microscopic features. However, we would like to point out that an end-to-end neural network trained *similar* to a human, *i.e.* shown videos of processes with various particle sizes and then classifying them, would still work like a black box. Therefore, its results would not be interpretable, neither would its generalization ability be quantifiable. The surface texture relies on system parameters such as the powder materials composition, solvent types, dryer shape and even impeller shape, and the operation would also become sensitive to lighting orientation and other operational parameters. This is one challenge that computer vision researchers are very familiar with from other applications (e.g. face recognition). Our method relates microscopic powder features to the PSD by relying on optical physics, which is insensitive to the unit operation parameters, and yields better repeatability, generalization ability and interpretability. The optical physics-based model also helps us reduce the amount of data required for efficient training.

Particle size of 700 micron is large. There are many objectives with appropriate magnification to detect particles over 100 micron. I also work with KCl and I have photos about KCl crystals ($DV_{10} = 26$ micron, $DV_{50} = 186$ micron, $DV_{90} = 402$ micron) with a 35

mm focal length objective (https://pyramidimaging.com/Basler-Lens-C11-3520-12M-P-f35mm---Lens_p-73256.aspx). (Working distance was about 150-200 mm).

We agree that computer vision with a long working distance objective should work in principle, but we would like to point out that the objective lens has fundamental limits in some industrial-scale dryers while our method is deployable to many different drying systems.

First, a working distance about 150-200mm is not long enough for a big dryer, in which the distance from the window to powder surface could reach 0.5-1m. Under this paragraph, we attach a specification sheet from De Dietrich as an example of the dimensions. Our laser speckle probe does not face this challenge since we illuminate the powder surface with a collimated beam, thus no focusing is required. The coherent laser beam easily maintains its collimation for over 2 meters propagation distance, and this guarantees the applicability of our method in such large-scale dryers. Moreover, since agitation could always change the powder surface height, the objective lens might either go out of focus or require additional auto-focusing computations which might reduce its robustness. Our method is insensitive to the distance from the window to the powder surface. Last but not least, our method maintains the option of detecting smaller particle sizes, e.g. 5-200 μm (as discussed in the Supplementary section 5), should other applications require it. Directly imaging these small sizes, while the particles are being violently agitated, with an objective at 10+ cm distance

would be rather challenging.

Redacted

3. Does it work with needle-like crystals as well? Many types of aggregation can occur. It is very time consuming to “teach” new models for the all the new materials. Aggregated particles also should be shown to the model during the learning phase as well.

We appreciate the Reviewer steering us toward an interesting test of generalization ability for our method. We have validated the answer from both the theoretical model in the original manuscript and a new experiment which we have added for this revision.

The application for the non-spherical shape particles was introduced in the “discussion” section and more discussion is in the Supplementary section 4, line 218. *“Moreover, this physical meaning applies to any particle shape besides round particles, it is easily generalized to the rotationally averaged cumulative size distributions for non-rotationally symmetric cases, such as the cubic-shaped KCl powder particles shown in Fig.2a.”* We defined the particle size

distribution for non-spherical particles as the rotationally averaged size distribution along all directions, based on our physics model. With this physics definition, our inverse model can still maintain good generalization ability in the purpose of size detection, although it is not trained by different powder morphologies or aggregate shapes.

Fig. S7 | Validation results for the needle shape powders L-threonine (i) and Monosodium glutamate, MSG (ii). (a) In the microscopic images with the same scale bar for both (i) and (ii), MSG is clearly much bigger than L-threonine. (b) Measured averaged autocorrelations showing that MSG results in weaker side lobes compared to L-threonine. (c) Particle size distributions in volume base measured by our method (orange bar) and the Mastersizer (red line) serving as the ground truth.

“To validate the performance for needle shape particles, we trained the pipeline with speckle from sieved KCl particles and tested it with L-threonine and Monosodium glutamate (MSG). The result is shown in Fig. S7. The microscope images in (a) show visually clear size difference between these two powders, and verifies that they are all needle shaped. Part (b) shows averaged autocorrelations. According to Fig.2g in the main text, the first order side lobe is merged into the main lobe and is hard to resolve. L-threonine has a stronger second order side lobe than MSG. Higher order side lobes cannot be observed for either material. The particle size distributions transformed into volume base are plotted in part (c). The L-threonine’s speckle prediction matches the Mastersizer result. The prediction for MSG has the same peak position with the ground truth but different widths. We may explain this width mismatch from two viewpoints. The first one is that

the ground truth size distribution disperses more than 1000 μ m (1mm), which is out of the range in our training sets because KCl never forms particles as large. The second reason is that our model cannot work with the bimodal distribution very well, as we mention in the Discussion section of the main text and further discuss in Supplementary Section 6. This experiment confirms that to certain extent our model can apply to different particle shapes without retraining the neural network.” Interpretability and generalization ability are two of the main advantages for the physics enhanced machine learning algorithm, as we also pointed out in response to the Reviewer’s earlier question *vis-à-vis* computer vision.

The above paragraph related to Fig. S7 has also been added to the Supplementary Section 4. The following sentence has been added to the Discussion section in the main text, starting at line 183.

“The output of the estimator is the cumulative size distribution, which is applicable to different particle shapes. Test results on needle shape materials, along with a theoretical discussion of the relationship between non-spherical particles and cumulative distributions are included in Supplementary Section 4.”

4. If the crystal morphology is changed during the crystallization the laser speckle will change as well.

We thank the Reviewer for raising the potential challenge to our method. We would like to point out that in pharmaceutical industry practice crystallization is a different unit operation than drying. The focus of our present work is to track the particle size distribution during drying. In such a case, from known process knowledge, for example that dehydration could break the crystals but has only little chance to change their morphology, we could safely rule out the possibility of the laser speckle pattern’s statistics being affected by changes in the particulates’ morphology. Moreover, as discussed in our response to question 3, our method would be compatible for different morphologies because it generalizes well. Thus, it should still be able to provide correct PSD estimates even if the morphology changes unpredictably for some reason, within the bounds delineated in our manuscript.

Reviewer #2 (Remarks to the Author):

Zhang et al. describe a novel quantitative analysis method that uses back-scattered speckle images to estimate particle size distribution (PSD). This method can be applied to monitor powder size distribution in the pharmaceutical industry. In the analysis, a physical forward model based on scattered field data is used together with partial experimental data in a deep neural network to generate further synthetic data and estimate PSD with an estimator to solve the inverse problem. In general, the research problem in the paper is interesting and solid, in the sense that extracting quantitative information from scattered fields is not an easy task in two aspects: understanding the image formation process and solving parameter estimation inverse problems in a generalized way that is applicable to real-life situations. In order to solve both aspects of the problem, the authors incorporated physics-informed machine learning. In addition, the paper is overall carefully and rigorously written. I particularly appreciate the way in which the limitations and weaknesses of the method are explained clearly and honestly. However, I still have several major concerns regarding the paper that need to be addressed before I consider it for acceptance:

We thank the Reviewer for the encouraging overall opinion and for the detailed criticisms below. Our responses are as follows.

Major comments:

1) An important application of the work is the monitoring of the wet powder drying process in a "real-time, non-invasive" manner. The data and setup demonstrated so far only support the non-invasive claim. The real-time claim, which should be one of the most exciting claims, does not appear to be supported by any obvious data:

1.1) During the drying of wet powder, I would expect dynamic variations in speckle images that may decorrelate over time, so the ensemble average model may not be applicable. There is a need for the authors to provide more technical details regarding the imaging and data

processing time in comparison with the speckle decorrelation time in the typical wet powder drying process. To support the real-time claim, I would expect a time-lapsed measurement of PSD.

We appreciate the Reviewer bringing up this important point about the speckle decorrelation. This is one of the most salient aspects of the implementation of this technique. We agree that speckle will decorrelate over time. However, we would also like to point out that our measurement is not limited by the decorrelation for the following reasons. (In addition, a time-lapse PSD measurement which partially addresses this point is discussed in our response to question 1.3).

(1) To validate that our exposure time is short enough to maintain the speckle spatial correlation within each frame, we collected single frame images with different exposure times. The raw single-frame speckle images and the corresponding autocorrelations are attached in Fig. S2. 100 μs and 200 μs exposure times still maintain their spatial correlation, and they begin to decorrelate from 500 μs onwards. The speckle pattern is blurred when exposure becomes as high as 1ms whence the corresponding autocorrelation disperses. These data convince us that our single frame exposure time 100 μs is sufficiently smaller than the speckle decorrelation time in a typical powder drying process.

We have added Fig. S2 and modified the description about the exposure time in the Supplementary Section 1 as in the following quotation.

Fig. S2 | Speckle images (top) and its autocorrelations (bottom) in different exposure times.

Original:

“The exposure time is 100μs, fast enough to avoid temporal blurring of the speckle due to the agitation.”

Modified:

“Fig.S2 shows the single-frame raw speckle images and the corresponding autocorrelations with different exposure times. We find that 100 μs and 200 μs exposure times maintain high degree of spatial correlation whereas the speckle begins to decorrelate at exposures in excess of 500 μs. At exposure as high as 1ms the speckle pattern blurs and the corresponding autocorrelation disperses. We choose 100μs as our exposure time to ensure that it is short enough to maintain the speckle spatial correlation within each frame.”

(2) In our drying process, the impeller rotates at most at 4 rpm (as mentioned in line 208) which corresponds to the largest speed of 4.2cm/s. Within the 100μs exposure time, the longest impeller motion is only around 4μm, which is much smaller than the particle size (50-1000μm). Moreover, the powder motion speed is much smaller than the impeller’s speed throughout most of the process, except at the very beginning when the viscous force causes all the particles to adhere and move in tandem with the impeller. Within our time window of

interest, which is generally after this initial stage has ended and normal mixing is taking place, the powder motion in 100 μ s exposure time is typically smaller than the wavelength (validated by the above-mentioned experiment), which ensures that the spatial correlation of the speckle pattern is maintained.

(3) According to the ensemble averaging operation in our forward model, temporal correlation among different frames is not required. In fact, the opposite is the case: we must make them temporally decorrelated to ensure statistical independence among frames. In our measurement, the time step between frames is 14ms (frame rate is 70 fps, mentioned in Sup. Line 46), which is much larger than the decorrelation time. Thus, we ensure the integrity of our ensemble average operation.

1.2) It also appears that the model does not account for dynamics as well. Does the model assume that each measurement is made within the period of speckle correlation?

We thank the Reviewer for this point. Our approach is subject to dynamics in the system at two different time scales. The first is the speckle dynamics, with a time scale of hundreds of microseconds. We have taken it into account as discussed in our response point (2) to the Reviewer's question (1.1) above. The second time scale is the dynamics of the powder drying itself, which is tens of minutes or even longer (as shown in Fig. 5 addressed in question 1.3). However, it takes only 15 seconds to provide a PSD measurement, including data collection time for 200 frames and the computation time. The computer used for this measurement has Intel Xeon W2245 CPU, 64 GB RAM, and NVIDIA Quadro RTX 5000 GPU with 64 GB VRAM. Since the time scale of the drying dynamics is much longer than the PSD measurement period., we can safely assume that the powder PSD will not change during the data collection time for the temporal averaging.

We have rephrased the related statement in Supplementary Section 1 as follows.

Original: *“Since the dynamics of agglomeration is much slower (typically is from tens of minutes to hours) than this time scale, it is safe to assume the PSD remains the same during the*

data collection time.”

Rephrased: *“The time scale of PSD evolution in the powder drying dynamics is tens of minutes or even longer (as shown in Fig.5) which is much longer than the time of a single PSD measurement. So we can safely assume that the powder PSD does not change during the data collection time.”*

1.3) Although the authors provided a ground truth comparison study, it was done with fixed powder rather than wet drying powder as claimed in the introduction. Is there any experiment to validate the "real-time" measurement component? It is necessary for the authors to provide more image data from different times, as well as its analysis during the validation experiment with a known ground truth PSD. Otherwise, this claim should be tuned down and another discussion section should be added to the paper.

We appreciate both Reviewers prompting us to investigate “real-time” measurements. We carried out a time-lapse experiment for a demo drying process, shown in Figure 5 (the same experiment was used to address some of the questions of Reviewer 1). Fig.5b shows a PSD map versus time, and PSD curves sampled at different times are plotted in Fig.5a. The sampling period of the PSD measurement is 15 seconds, including data collection time for 200 frames and the computation time. We also placed Mastersizer results as the ground truth next to the laser speckle predictions at the beginning and ending times. We did not measure the PSD in the process with Mastersizer because the soft agglomeration in this demo process is too fragile to apply any mechanical contact. Parenthetically, this is one of the advantages of our non-invasive measurement compared to the traditional characterization methods. Fig. 5c are camera photos for reference, captured with an iPhone 11, corresponding to the marked times in (a) and (b).

Fig. 5 and the details about this experiment are added to the manuscript as quoted below.

Fig. 5 | Time-lapse PSD measurement in the drying process. (a) Measured PSDs at a few selected time points corresponding to the dashed red line marks in (b). The dashed line plots are the Mastersizer measurements serving as ground truth at the beginning and the end of the process. (b) Time-lapse PSD map during the drying process. We observed soft agglomeration occurring from 5-25 min, and deagglomeration from 25-40 min. (c) Corresponding camera photos at the times where the PSDs were sampled in (a). These photos were taken from the same optical window as the laser beam using a commercial iPhone 11.

“To validate the real-time monitoring applicability of our method, we carried out a time-lapse PSD measurement of an entire filter drying process⁴¹. The detailed information of our dryer is included in Supplementary Section 1. This demo process operated on 280g of KCl powder with a mixed solvent (water 40g/Ethanol 60g). Throughout the process, we maintained conditions of temperature 26 °C, pressure -720 mbar and agitation speed 4 rpm. Fig. 5b shows the PSD map vs. time. The sampling period of the PSD measurement was 15 seconds, including the data collection time for 200 frames and the computation time. The computer used for this measurement was an Intel Xeon W2245 CPU, 64 GB RAM, and NVIDIA Quadro RTX 5000 GPU with 64 GB VRAM.

For the duration 5 to 25 minutes since the beginning of the process, we observed a gradual size increase compared to the original PSD. Since no crystallization or crystal growth could take place during drying, we think soft agglomeration instead occurred. From 25 to 40 minutes, the PSD gradually decreased to the original distribution, which is indicative of deagglomeration. At the far

end (>75 mins), crystal breakage⁴² caused the size distribution to slightly decrease compared to the original. PSD curves sampled at different times are plotted in Fig. 5a, together with Mastersizer results serving as ground truth for the beginning and ending time. They match well and the slight crystal breakage is clearly resolved for both Mastersizer and speckle predictions. The PSD curve shifts toward the right from the start time to 26.1 minutes, and then decreases backwards to the original position as plotted for the next 59.2 min. We did not measure the PSD with the Mastersizer when the powder was wet because in this process soft agglomerates were too fragile to allow any mechanical contact. This is one of the advantages of our non-invasive measurement compared to the traditional way with the Mastersizer. Fig. 5c are camera photos, taken with an iPhone 11, corresponding to the marked times in (a) and (b) for reference. (i) and (iv) are dry powders, while the (ii) and (iii) are wet powder images. It is hard to distinguish different sizes from the macroscopic textures.”

2) As the data analysis method in the paper has only quantified particle sizes of a few hundred microns, which is within the capability of a typical wide-field microscope with a large field of view design with a few hundred microns of resolution. What is the clear technical advantage of this method over direct microscopic imaging?

We thank the Reviewer for this comment. The comparison between our method and the microscope imaging with a long working distance objective lens helps us articulate the differences between our method and more traditional ones, based on computer vision; and it had been indeed the subject of long internal discussions among our team during the formulation stage for this project.

Using objective lenses faces certain practical limitations in industrial-scale dryers. The first is the working distance. As Reviewer #1 pointed out, and we agree, it is possible to directly image fairly large particles, about 0.5mm or larger, with an objective lens placed at about 15-20cm away from the drying surface. To image smaller particles, the objective would have to come closer, adding to the challenge. Even working distances of 15+ cm are not long enough for a big dryer, where the distance from the window to powder surface could exceed 0.5-1m.

Under this paragraph, we attach a specification sheet from De Dietrich as an example of the dimensions. Our laser speckle probe doesn't have this limitation since we illuminate the surface with a collimated beam and no focusing is required. The coherent laser beam easily maintains its collimation over 2m or more of free space propagation. Moreover, agitation could cause the powder surface to swell or recede within the field of view, driving the objective out of focus. Speckle is an interference-based effect, which renders our approach insensitive to defocus. Last but not the least, our method keeps open the option of detecting particles of smaller ranges (as discussed in the Sup, section 5), e.g. down to 5-200 μ m.

Redacted

3) Codes for the model and DNN should be publicly available for easy reproducibility of the work.

We thank the Reviewer and absolutely agree with the principle of data sharing. We have posted the code in GitHub https://github.com/qhzhang95/PEACE_Speckle and the data is available on <https://dataverse.harvard.edu/dataset.xhtml?persistentId=doi:10.7910/DVN/FZUG9V>. We have

also updated the “data and code availability” section in the manuscript after the Method section.

“

Data Availability

The data generated in this study are provided in the Harvard Dataverse under <https://dataverse.harvard.edu/dataset.xhtml?persistentId=doi:10.7910/DVN/FZUG9V>.

Code Availability

All original codes have been deposited at https://github.com/qhzhang95/PEACE_Speckle and is publicly available.

”

4) In lines 38-39, "The laser speckle pattern has long been used to characterize surface roughness. ", but there are not enough references cited and attributed, please cite the work Buchanan, J., Cowburn, R., Jausovec, AV. et al. Nature 436, 475 (2005) and other significant previous work.

We thank the Reviewer for bringing up this reference, hitherto unknown to us. It is inspiring to learn that laser speckle resolves the surface imperfections in documents and packaging as a physical identity code. Our technique is indeed potentially applicable to this problem as well. More related works cited in our revised manuscript are shown below.

4. Buchanan, J. D. R. *et al.* ‘Fingerprinting’ documents and packaging. *Nature* **436**, 475 (2005).
5. Yan, J. *et al.* Recognition of Suspension Liquid Based on Speckle Patterns Using Deep Learning. *IEEE Photonics J.* **13**, 1–7 (2021).
6. Kalyzhner, Z., Levitas, O., Kalichman, F., Jacobson, R. & Zalevsky, Z. Photonic human identification based on deep learning of back scattered laser speckle patterns. *Opt. Express* **27**, 36002–36010 (2019).

7. Dogan, M. D., Acevedo Colon, S. V., Sinha, V., Ak\csit, K. & Mueller, S. SensiCut: Material-Aware Laser Cutting Using Speckle Sensing and Deep Learning. in *The 34th Annual ACM Symposium on User Interface Software and Technology* 24–38 (Association for Computing Machinery, 2021).
8. Valent, E. & Silberberg, Y. Scatterer recognition via analysis of speckle patterns. *Optica* **5**, 204–207 (2018).
9. Lotay, A., Buttenschoen, K.-K. K. & Girkin, J. M. Quantification of skin quality through speckle analysis. in *Photonic Therapeutics and Diagnostics XI* (eds. Kang, H. W. et al). vol. 9303 93030Q (SPIE, 2015).
10. Cozzella, L., Simonetti, C. & Schirripa Spagnolo, G. Drug packaging security by means of white-light speckle. *Opt. Lasers Eng.* **50**, 1359–1371 (2012).
11. Rey-Barroso, L. *et al.* Optical Technologies for the Improvement of Skin Cancer Diagnosis: A Review. *Sensors* **21**, (2021).
12. Bar, C., Alterman, M., Gkioulekas, L. & Levin, A. Single scattering modeling of speckle correlation. in *2021 IEEE International Conference on Computational Photography (ICCP)* 1–16 (2021). doi:10.1109/ICCP51581.2021.9466262.

Minor comments:

1) *According to equation M15, is the ensemble average the average over time? Please clarify in notation which parameter in $A(u)$ is averaged from equation M14 to M15. If it is time or the number of measurements, then M14 should include the parameter t , i.e. If it is time or the number of measurements, then M15 should be $A(u)=|t$. In the paper, it is also not stated clearly how many average times have been calculated for each experiment. Please clarify.*

We Reviewer’s thorough review of our derivations. In equation M15,

$$\begin{aligned}
\langle A(u) \rangle &= \int d\tau W(\tau) \left\langle \left| \sum_i e^{-ju(x_i + \frac{\tau}{2})} \frac{\sin\left(u\left(r_i - \frac{\tau}{2}\right)\right)}{u} \right|^2 \right\rangle \\
&= \int d\tau W(\tau) \int p(r_1)p(r_2) \frac{\sin\left(u\left(r_1 - \frac{\tau}{2}\right)\right)}{u} \frac{\sin\left(u\left(r_2 - \frac{\tau}{2}\right)\right)}{u} dr_1 dr_2 \left\langle \sum_{i,j} e^{-ju(x_i - x_j)} \right\rangle
\end{aligned} \tag{M15}$$

the ensemble average is over independent measurements, as we pointed out in response to question (1.1) part (3). Specifically, it is an average over x_i . However, x_i is a function of time t in our system since the powder are agitated by the impeller. We have substituted x_i with $x_i(t)$ as the Reviewer suggested. Since $x_i(t)$ is ergodic, we may replace the average of measurements at different time t with the ensemble average. The 200 (or 1000) frames that averaging takes place over were specified in the main text (line 133) and the caption of Figures 2 and 3, respectively.

Equations M14 and M15 and the related descriptions have been revised as in the following quote:

“

$$A(u, t) = \left| \sum_i \frac{\sin(r_i u)}{u} e^{j2\pi x_i(t)u} \right|^2 . \tag{M14}$$

Here, $u = \frac{u'}{\lambda f_3}$, i is the index of the i -th particle, and r_i and x_i are the radius and the position for the i -th particle, respectively. This is the same equation (5) in the main text.

Below are the derivations of the ensemble average of the autocorrelation $A(u, t)$. *The ensemble average is over independent measurements. Specifically, it is an average over x_i . However, x_i is a function of time t in our system since the powder are agitated by the impeller. Since $x_i(t)$ is ergodic, we may replace the average of measurements at different time t with the ensemble average. Starting from equation (M14), and since $W(\tau)$ is invariant, we may move the ensemble average bracket into the integral, as*

$$\begin{aligned}
\langle A(u) \rangle_t &= \int d\tau W(\tau) \left\langle \left| \sum_i e^{-ju(x_i(t) + \frac{\tau}{2})} \frac{\sin\left(u\left(r_i - \frac{\tau}{2}\right)\right)}{u} \right|^2 \right\rangle_t \\
&= \int d\tau W(\tau) \left\langle \sum_{i,j} e^{-ju[(x_i(t) + \frac{\tau}{2}) - (x_j(t) + \frac{\tau}{2})]} \frac{\sin\left(u\left(r_i - \frac{\tau}{2}\right)\right)}{u} \frac{\sin\left(u\left(r_j - \frac{\tau}{2}\right)\right)}{u} \right\rangle_t \\
&= \int d\tau W(\tau) \int p(r_1)p(r_2) \frac{\sin\left(u\left(r_1 - \frac{\tau}{2}\right)\right)}{u} \frac{\sin\left(u\left(r_2 - \frac{\tau}{2}\right)\right)}{u} dr_1 dr_2 \left\langle \sum_{i,j} e^{-ju(x_i(t) - x_j(t))} \right\rangle_t
\end{aligned} \tag{M15}$$

”

2) *The setup description in the Method part should include more information, such as the focal lengths of the lenses, camera information, etc.*

We appreciate the Reviewer bringing up this point. The focal lengths of the lenses were included in the caption of the extended data of Fig 1. The camera information and other optics details were introduced in the Supplementary section. 1.

To make things clearer to the readers, we have rephrased the corresponding passage in the Methods section as:

“The optical beam path is shown in Extended Data Fig. 1b. The laser beam is expanded to 4.8 mm with a beam expander in the telescope configuration, consisting of lenses L1 (focal length 25mm) and L2 (30mm). The beam is initially polarized in the direction parallel to the plane shown in the diagram, and so it is reflected by the polarizing beam splitter (PBS) to reach the sample inside the dryer through the glass window. Passage twice through the quarter-wave plate rotates the outgoing polarization direction from parallel to perpendicular so that the scattered light from the potassium chloride (KCl) powder sample is now transmitted through the PBS and propagates vertically upwards. The wave plate is also tilted by approximately 10 degrees for the same reason as the beam, to minimize specular back-reflection. Lens L3 (250mm) concentrates the scattered light so that an angular range that is as large as possible can be captured by the CCD (model ZWO ASI183MM Pro). More information about our apparatus can be found in Supplementary Section 1.”

REVIEWERS' COMMENTS

Reviewer #1 (Remarks to the Author):

I accept the answers of the authors.

Reviewer #2 (Remarks to the Author):

My previous comments have been satisfactorily addressed by the authors. I don't have any more comments. The current manuscript has been greatly improved for publication.